# Towards a Neurophenomenological Understanding of Self-Disorder in Schizophrenia Spectrum Disorders: A Systematic Review and Synthesis of Anatomical, Physiological, and Neurocognitive Findings

**DOI:** 10.3390/brainsci13060845

**Published:** 2023-05-23

**Authors:** James C. Martin, Scott R. Clark, K. Oliver Schubert

**Affiliations:** 1Discipline of Psychiatry, Adelaide Medical School, The University of Adelaide, Adelaide, SA 5000, Australia; scott.clark@adelaide.edu.au (S.R.C.); oliver.schubert@adelaide.edu.au (K.O.S.); 2Basil Hetzel Institute, Woodville, SA 5011, Australia; 3Division of Mental Health, Northern Adelaide Local Health Network, SA Health, Adelaide, SA 5000, Australia; 4Headspace Early Psychosis, Sonder, Adelaide, SA 5000, Australia

**Keywords:** self-disorder, anomalous self-experience, schizophrenia, schizotypy, ipseity, perceptual disintegration, sensory attenuation, default mode network

## Abstract

The concept of anomalous self-experience, also termed Self-Disorder, has attracted both clinical and research interest, as empirical studies suggest such experiences specifically aggregate in and are a core feature of schizophrenia spectrum disorders. A comprehensive neurophenomenological understanding of Self-Disorder may improve diagnostic and therapeutic practice. This systematic review aims to evaluate anatomical, physiological, and neurocognitive correlates of Self-Disorder (SD), considered a core feature of Schizophrenia Spectrum Disorders (SSDs), towards developing a neurophenomenological understanding. A search of the PubMed database retrieved 285 articles, which were evaluated for inclusion using PRISMA guidelines. Non-experimental studies, studies with no validated measure of Self-Disorder, or those with no physiological variable were excluded. In total, 21 articles were included in the review. Findings may be interpreted in the context of triple-network theory and support a core dysfunction of signal integration within two anatomical components of the Salience Network (SN), the anterior insula and dorsal anterior cingulate cortex, which may mediate connectivity across both the Default Mode Network (DMN) and Fronto-Parietal Network (FPN). We propose a theoretical Triple-Network Model of Self-Disorder characterized by increased connectivity between the Salience Network (SN) and the DMN, increased connectivity between the SN and FPN, decreased connectivity between the DMN and FPN, and increased connectivity within both the DMN and FPN. We go on to describe translational opportunities for clinical practice and provide suggestions for future research.

## 1. Introduction

For decades, clinical psychiatry has operated within categorical classifications of mental disorders, formed from clusters of phenomenological features (syndromes) [1]. The evidence base for treatments, including pharmacology, psychotherapy, and functional interventions, arises from clinical trials and is manifest in resultant treatment guidelines built upon these diagnostic constructs. Standardized use of syndromal psychiatric diagnoses, without reference to underlying neurobiological correlates, contributes to high heterogeneity within diagnostic groups in both phenotype and treatment response, as well as high comorbidity between disorders and arbitrary boundaries between normal and abnormal [2]. Across disorders, there are high levels of treatment resistance [3] and generally modest effect sizes for pharmacological and psychological treatments [4,5].

Furthermore, diagnostic classification systems, such as the DSM-5TR and ICD-11 [1,6], have sacrificed detail and completeness for standardization of simplified concepts and have progressed from their original intention, as indicators of complex conditions, to “defining” the diseases themselves [7,8]. For example, whilst schizophrenia is currently defined only by the presence of psychotic features such as hallucinations, delusions, and disorganized or negative symptoms [1], these symptoms were historically considered peripheral; its core was, instead, best characterized by a loss of the innermost self [9]. In modern psychiatry, whilst clinicians are still taught to identify relevant phenomena as part of a mental state examination, such as passivity phenomena where one believes one’s thoughts or actions are controlled externally, constructs such as the ‘self’ are not well operationalized in current diagnostic criteria and exploration of phenomenological experience tends to focus almost exclusively on the content of experience [1,10,11]. In contrast, Sterzer and Mishara [11] (p. 6) propose greater progress will come from “*not the content of the thoughts, volitions, etc., but the manner of their givenness, not the ‘what’ but the ‘how’ of the experience*”. For example, what makes passivity symptoms aberrant is less likely to involve a person’s attribution of the movement of their limb, such as ‘my arm was moved by the nurse’, but rather the perceptual experience itself [12], ‘my experience of moving my arm feels radically altered in my awareness’.

These considerations urge further exploration and synthesis [13,14] in order to provide comprehensive diagnostic and explanatory models that more closely reflect first-person experience whilst recognizing their reciprocal relationship to underlying neurophysiology [15]. Such a neurophenomenological approach [15] may provide improved understanding of underlying mechanisms of psychosis leading to novel avenues for treatment and improve shared understanding of patient experience promoting better engagement and adherence to treatment. In this systematic review, we aim to provide an overview and synthesis of one more recent phenomenological concept capturing the ‘how’ of patients’ experience, that of Self-Disorder (SD) or disordered ‘minimal self’ [16], in schizophrenia spectrum disorders (SSDs).

Self-Disorders (SD) are involuntary subjective disturbances of the ‘given’ experience of ‘minimal self’. ‘Minimal self’ experience involves a sense of ‘mineness’, or ‘ipseity’, of embodied experience. When ‘minimal self’ experience is disturbed, patients might report feeling *as if* they are detached from reality, are devoid of agency, or are a passenger in their body and mind. Ipseity disturbance is hypothesized to result from the interactions of three features or processes: *Hyper-reflexivity*, *diminished self-presence*, and ‘*disturbed grip’* [17]. *Hyper-reflexivity* refers to an enhanced inward awareness, an involuntary conscious awareness of core aspects of ‘minimal self’ experience that usually remain unconscious, transparent and un-questioned. The process of hyper-reflexivity then generates an externalization of ‘minimal self’ experience, leading intrinsic signals such as one’s heartbeat to be perceived as external to the self. *Diminished self-presence* pertains to a diminished experience of existing as a source of awareness or an entity with the capacity to act on the environment, for example wondering whether one truly exists. The final component is a ‘*disturbed grip*’, a loss of salience or meaningfulness of stimuli in the field of awareness [16,17], such as feeling lost or distant from one’s body, limbs, or even one’s thoughts. The phrase *as if* is often a distinguishing feature when patients describe SD, highlighting the difficulty patients have when effectively formulating and communicating the nature of these experiences. Characteristics of SD also overlap with Anomalous Self Experience (ASE) and Depersonalization Disorder (DPD) [18], where all three terms have been used synonymously in research, despite common taxonomic conceptualizations of DPD as an anxiety disorder [1]. Yet, features of SD, such as a loss of agency, are considered more typical in SSDs, not anxiety disorders [19]. Further, transient disturbances in ‘minimal self’ awareness are common across both clinical and non-clinical populations. Examples range from the heightened awareness of gustatory function in those with poor interoceptive ability [20,21] to the introspective experiences associated with advanced mindfulness [12]. In fact, abnormal self-experiences can be induced experimentally using paradigms such as the false hand illusion [22]. Yet, unlike these experiences, Self-Disorder is not transient, nor does it tend to remit without intervention. Crucially, insight is a key feature that differentiates psychotic from neurotic states, and patients with SD may be present in both clinical groups, yet those with insight remain aware that their experience is not reality-based. Herein, we use the term Self-Disorder to encompass all subjective disturbances of ownership, agency, and vitality. We also integrate and discuss SD within the psychosis spectrum.

Prominent psychopathologists have long advocated for self-disturbance to be considered a core feature of SSDs [12,16]. In support of these claims, SD shows strong temporal stability [23] and is present in many patients before and after episodes of acute positive psychotic symptoms such as delusions or hallucinations. Two recent meta-analyses further concluded that SD hyper-aggregates in SSDs but not in other mental illnesses nor controls; they are approximately 4.5 times more likely to occur in SSDs than non-SSD populations [19,24]. The authors go on to implicate SD as an indicator of vulnerability for psychosis, suggesting a generative relationship between SD and later onset of psychotic features [19]. Yet, until recently, SD was neglected in research settings, and continues to be so in most clinical contexts, thus leaving many patients without an adequate model of their experience.

Two neuropsychological models of SD have been experimentally tested and can inform a neurophenomenological approach: the Ipseity Disturbance model and the Active Inference model. The Ipseity Disturbance model (‘Basic Self Model’) rests on the assumption of a ‘minimal’ or ‘pre-reflective’ self, and a ‘first-personness’ or ‘ipseity’ of experience, “*the core sense of existing as the subject of one’s own experience and agent of one’s own actions*” [17] (p. 720). The Ipseity Disturbance model aims to bridge gaps between phenomenology and cognitive science by implicating two cognitive features, *Aberrant Salience* and *Source Monitoring* to explain patient experience. Aberrant salience refers to an impaired ability to distinguish between salient and non-salient stimuli. Source monitoring (reality monitoring) deficits refer to difficulties inferring the source of sensory signals between self and other. The model also argues for an additional ‘trait-like’ component of ‘*perceptual disintegration*’, which is thought of as an impairment of multi-sensory integration. Using this model, two measures of SD were developed and have been validated extensively. The Examination of Anomalous Self-Experience (EASE) is a semi-structured interview that assesses ipseity disturbance across five dimensions: stream of consciousness, sense of presence, sense of corporeality, self-demarcation, and existential reorientation [25]. The Inventory of Psychotic-like Anomalous Self-Experiences (IPASE) [26] is a 57-item self-report screener with a similar five-dimensional structure. The authors of the EASE identified a third measure—the Bonn Scale for the Assessment of Basic Symptoms (BSABS) [27,28], a 92-item semi-structured interview—which substantially overlaps with symptoms of SD [25], though it was not designed to specifically measure the construct.

A second model, Active Inference, operates through a Bayesian framework, which presupposes the brain as an organ of statistical inference, predicting current and future events based on past experience and sensory information [29,30]. A comprehensive description of Active Inference models falls outside the scope of this review but can be found elsewhere [31]. Broadly, Active Inference proposes that conscious beings exert control over their environment by engaging in reciprocal interactions through an action–perception loop [31], with the fundamental purpose of survival being ‘*preserving integrity by fighting entropy*’. As such, organisms act to minimize the variance between predictions [prior beliefs] and observations [sense data], a value often referred to as ‘*surprise*’ or ‘*free energy*’ [32]. When self-generated efferent signals are produced, such as in movement, thought, and even unconscious interoceptive signaling, Active Inference hypothesizes an ‘*efference copy*’ is also produced to inform the brain that the action is self-generated and to adjust perception accordingly. As these signals are highly predictable, we can afford to process them outside conscious awareness. Consequently, most people produce efference copies to suppress ‘*minimal self*’ signals [16,33], such as gustatory processes, heart rate, and somatosensory pathways. This allows the self to remain transparent and fully immersed with the world, whilst also allowing attentional resources to be directed towards more salient stimuli [18]. Disturbances in the production, relay, or receipt of efference copies would likely result in a failure of attenuation of self-generated signals, producing abnormal salience landscapes (i.e., unpredictable stimuli are pertinent to survival). Emerging empirical studies have begun to provide insight into the underlying biology, implicating mesolimbic dopamine pathways in the substantia nigra, as well as brain regions such as medial/lateral pre-frontal cortex, temporo-parietal junction, and the insular cortex [17,34], yet no framework currently exists to incorporate all of these findings.

In modern neuroscience, many methods can explore the physiological disturbances associated with SD, each of which may provide insight at separate levels of analysis. For example, blood biomarkers and genomic profiling can enhance our knowledge of how gene expression, protein development, neuronal development, or signal transmission may affect vulnerability or lead to symptom onset and remission. Concurrently, non-invasive procedures such as Electroencephalography (EEG) and Magnetic Resonance Imaging (MRI) may reveal structural or functional changes both within and across brain networks which manifest in altered self-perception [35,36,37], whereas Electromyography (EMG) and neurocognitive testing are able to quantify cognitive and behavioral abnormalities that may be associated with brain network dysfunction. Further investigation of such a diverse and complex construct, that of SD, requires analysis of progress already achieved. We aim to integrate current knowledge, aligning existing models of SD with the broader neurobiological and computational literature. Consequently, we provide a systematic review of the proposed associations between SD and neurophysiological and neurocognitive correlates.

## 2. Methods

### 2.1. Search Strategy

JCM searched the PubMed database for peer reviewed articles, using the following search terms: (EASE OR IPASE OR BSABS) AND (schizophrenia OR psychosis OR schizotypy OR dissociative disorders OR depersonalization) AND (EEG OR imaging OR cognition OR default mode network OR salience network OR executive function OR serum OR trauma OR personality). Articles retrieved from this search were subsequently screened by JCM, using PRISMA guidelines, for inclusion and exclusion criteria, first by abstract, then by full text (see Figure 1).

### 2.2. Inclusion Criteria

Consideration of SD and neurophysiological and neurocognitive endophenotypes.

### 2.3. Exclusion Criteria

Absence of a validated measure of Self-Disorder with demonstrated α-coefficient > 0.50, which is deemed acceptable according to Taber [38];Absence of a physiological measure as an experimental variable;Single-subject design/nonexperimental design.

Final decision to include or exclude papers where criteria were uncertain was made in conjunction with KOS and SRC.

The initial PubMed search strategy yielded 285 articles. Duplicate articles and those not in the English language were not found. We then commenced a full abstract screening using inclusion/exclusion criteria

## 3. Results

In total, 21 articles were included in the review (see Table 1, Table 2 and Table 3). Thematically, tables were separated according to physiological variables and neurological function: studies describing associations between SD and neural structure or function, studies describing associations between SD and perception, and studies describing associations between SD and cognition. All studies are data driven and, as shown in the tables below, there is considerable overlap, with many studies investigating measures of neural structure/function also providing important data on measures of cognition. In the sections that follow, we describe the main findings identified by these three classifications:

### 3.1. Studies Describing the Association between Measures of Self-Disorder and Physiological Measures Related to Neural Structure or Function

The included studies in Table 1 provide information gathered using a range of neurophysiological methods. Overall, these findings provide insight into the structural and functional abnormalities associated with SD. Firstly, two studies using Magnetic Resonance Imaging provided data on brain structure [39,40]. Bonoldi and Allen [40] showed the presence of a significant negative relationship between SD and grey matter volume within the anterior cingulate cortex, an area associated with self-referential thinking and mentalizing, though similar associations were not found in other midline structures implicated in the self, such as the posterior cingulate cortex and medial frontal gyrus [40]. Zhuo et al. [40] found no association between basic symptoms and midline cortical volumes.

Secondly, using either fMRI or EEG, seven studies found significant relationships between SD, or aspects of SD, and a range of abnormalities in brain network interaction, including both hyper- and hypoactivation. Respectively, these studies reported greater symptoms of SD were associated with hyperconnectivity within the Default Mode Network (DMN: medial prefrontal cortex, posterior cingulate cortex/precuneus and angular gyrus) [44], reduced connectivity between the dorsal anterior cingulate cortex and the Pre-Supplementary Motor Area [46], decreased perceptual organization [41], cortical hypersynchrony, defined as heightened connectivity across all cortical nodes [48], and lower EEG gamma frequency and higher peak beta amplitude over fronto-parietal regions in response to a proprioceptive stimulus [47]. Additional studies reported SD was linked to deficits within the magnocellular visual pathway [41,42] and a diminished ability to modulate EEG spectral entropy [48]. One further study compared EMG, a measure of muscular response, with BSABS scores, reporting a multi-modal dysfunction in emotional motor resonance characterized by a loss of emotional motor resonance in response to stimuli depicting positive emotions but excessive resonance to negative emotions [50].

### 3.2. Studies Describing the Association between Measures of Self-Disorder and Measures of Perception

We found two studies with significant findings linking SD to measures of perceptual aberration (Table 2). Composite scores for source monitoring, across both cognitive and EEG measures, explained substantial variance in SD (39.8%), and N1 suppression moderated this relationship in those with First-Episode Psychosis, but not in those at Ultra-High Risk of psychosis. Notably, measures of aberrant salience were not significantly associated with SD, suggesting impaired salience is not prerequisite to a disturbance of self [51]. Results of the second study indicate SD is associated with an impairment in the ability to recognize both positive and negative facial expressions, a task requiring the ability to infer the emotional state of others, often termed ‘mentalizing’ [52].

### 3.3. Studies Describing the Association between Measures of Self-Disorder and Measures of Cognition

Table 3 collates studies investigating associations between SD and different components of cognition [41,43,53,54,55,56,57]. At first glance, two patterns emerge. Firstly, there is evidence for a negative association of small-to-moderate effect size between cognitive scores and measures of SD across a wide variety of cognitive domains. Secondly, studies that use a less robust methodology by either utilizing a lower quality measure of SD, such as BSABS, in comparison to the EASE, or using composite scores of cognition, such as *General Intelligence*, are less likely to capture the nuances of the relationship between cognition and Self-Disorder, particularly in samples not experiencing acute psychosis.

## 4. Discussion

We aimed to provide a systematic review of reported associations between SD and neurophysiological and neurocognitive measures, towards a neurophenomenological approach to self-disturbance. In doing so, 21 articles were included in the review, and findings can be discussed across multiple levels of analysis. At the cellular and circuit level, SD is associated with deficits in M pathway processing, including M priming on the P pathway. At the network level, SD is linked to abnormalities across three distinct brain networks. Firstly, those with SD perform worse in several cognitive domains, including processing speed and executive functioning, both of which are associated with the Fronto-Parietal Network (FPN) [58,59], a coherent network of brain regions consisting of the lateral pre-frontal cortex, temporo-parietal junction, and dorsal/posterior parietal cortex [60] involved in attention, problem solving, and working memory [58]. Secondly, the Salience Network (SN) is formed from the anterior insula and the anterior cingulate cortex [61] and is believed to house the ‘minimal self’ [17], integrating multi-modal sensory signals, including interoceptive signals [17], and attributing salience to stimuli [60]. The detection of salient stimuli suggests the need to attend to cognitive demands; thereupon, the anterior insula activates the dorsal anterior cingulate cortex, which is believed to recruit the FPN. At the same time, the SN suppresses the third network, the Default Mode Network (DMN), made up of a range of midline brain regions such as the posterior cingulate cortex and medial pre-frontal cortex [59,61], which tend to activate in the absence of task requirements, and have been linked to self-monitoring, self-reflection, and self-concept [58]. Broadly, the findings of this review can be separated into themes of dysfunctional network connectivity, abnormal oscillatory activity, and multi-modal signal disintegration, which may contribute to the phenomenon of SD. Notably, these themes may represent distinct neural interactions, or different perspectives of the same process, as measured by different techniques. Below, we describe these themes in detail and then propose that they may be accommodated within a network model of brain dysfunction, informed by triple network theory [58].

### 4.1. Dysfunctional Brain Network Connectivity

Neural connectivity refers to activation patterns between distinct populations of neurons and can be measured in several ways. For example, fMRI can be used to map blood flow as an index of neuronal activity in specific brain regions, identifying networks which activate together under specific circumstances [36]. Patients displaying greater symptoms of SD displayed enhanced fMRI connectivity between one component of the SN, the right anterior cingulate cortex, and the right para-hippocampus, a region associated with the DMN. Similarly, enhanced intra-connectivity was reported between multiple additional brain regions associated with the DMN, including the right precuneus cortex, right para-hippocampus, and right isthmus cingulate cortex, all of which are associated with aspects of self-related cognition [44]. EEG can also be used to derive measures of connectivity through several analysis techniques, for example, the measurement of phase synchrony of brain regions defined as the connectivity strength between EEG nodes, the points of EEG signal transduction across the surface of the skull [62]. One study reported increased overall cortical connectivity in those with SD as well as a deficit in the ability to modulate overall cortical connectivity strength when comparing values pre and post an auditory oddball task [48]. A separate study found a diminished readiness potential in patients with SD [46]. Readiness potential is a slow brain wave which spikes a couple of seconds prior to self-generated movements, reflects a preparation for self-motion, and may be a critical component of self-awareness of movement. Effective connectivity between the Presupplementary Motor Area and the anterior cingulate cortex, a component of both the FPN and SN, is critical for the development of the readiness potential [46]. Similarly, the expectancy effect or ‘hazard function’ denotes how one’s expectation of a stimulus onset increases over time and can be measured by reaction time. In a trial where the target stimuli were purposefully not presented, those with symptoms of SD displayed a diminished expectancy effect [49], indicating an over-sensitivity to prediction error, where self-disordered patients are more likely to resolve uncertainty by relinquishing the state of expectation rather than enduring uncertainty. At the network level, this may be associated with decreased connectivity between regions of the FPN and DMN, an impairment in the ability to apply previous knowledge to current decision making, resulting in rapid shifting of beliefs [11,63]. Similarly, regions of the FPN, including the dorsolateral pre-frontal cortex, are believed to be the site of many cognitive processes, such as executive functioning, processing speed, and problem solving [58], impairment in each of which has been associated with greater symptoms of SD. Notably, these findings provide crucial insight into the many ways in which brain network dysfunction could occur.

### 4.2. Abnormal Oscillatory Activity

EEG waveforms are categorized within frequency bands that can be related to neurophysiological and cognitive processes [35]. This can be done either at rest, in the absence of any stimulus, or during a standardized task. Resting oscillatory activity represents spontaneous fluctuations in brain activity, which in alert healthy controls are often dominated by higher frequency bands such as alpha, beta, and gamma components [35]. Beta components oscillate at a lower frequency range (18–25 Hz) than gamma components (30–70 Hz), and EEG readings taken across frontal and parietal regions, when compared against EASE scores, displayed strong correlations across these bands in a small sample cohort of six schizophrenia and six schizotypal patients undergoing a proprioceptive task [47]. Specifically, lower-frequency gamma oscillations over parietal regions were linked with more severe SD, as were increased peak amplitudes of beta components over frontal and parietal regions [47]. Elsewhere, empirical findings in healthy controls indicate evoked gamma activity across fronto-parietal electrodes is linked to the perceptual aspects of a sensory stimulus [64] and its feedback (i.e., the nature and origin of stimuli) [65], whilst parietal event-related desynchronization in beta bands is seen during sensorimotor orienting tasks [66]. Consequently, the authors suggest SD may be characterized by a decoupling of motor and perceptual components with diminished processing of sensory feedback (i.e., diminished attenuation of self-signaling) and normal or enhanced processing of motor aspects [47].

### 4.3. Multi-Modal Signal Disintegration

Sensory signals are propagated through the nervous systems in distinct ways, dependent on modality. One example of such, the visual pathway, can be split into two separate components: the magnocellular pathway (M-pathway), which carries information about large, fast-moving objects, and the parvocellular pathway (P-pathway), which carries information about small, slow-moving objects. The M-pathway has been strongly linked to attentional modulation and salience attribution during visual processing, and both hyper- and hypofunction are associated with greater symptoms of SD [41]. The M-pathway also communicates with the P-pathway, termed M-priming, and those with SD display M-priming effects which hinder object recognition [42]. The authors argue increased variability of the M pathway (i.e., hyper- and hypoactivation) results in a decreased signal-to-noise ratio. They go on to suggest M pathway dysfunction may be the result of decreased temporal signal integration within early visual processing, though the specific mechanism underlying such dysfunction remains unclear.

### 4.4. Triple Network Theory as a Model of Psychopathology

Triple network theory prioritizes the role of the SN as mediating dynamic interactions between the DMN and FPN [58,67]. Triple network theory has been used to model a range of psychiatric disorders. In obsessive compulsive disorder, a hyperactivation of the SN and heightened connectivity with the FPN produces increased error monitoring, a hypersensitivity to negative feedback [68], and enhanced processing of interoceptive signals [69]. Similar findings have also emerged within neurodevelopmental disorders such as Autism and ADHD [70,71], mood disorders such as depression [72], and dissociative disorders [73], where derealization in particular has been associated with increased insula activation [73] and decreased DMN and FPN connectivity [74]. Furthermore, fMRI imaging research indicates hyperactive insula function causally underlies diminished connectivity between the DMN and FPN in schizophrenia patients, whilst hypoactive insula function enhances DMN and FPN inter-connectivity [75,76]. Specifically, a hyperactive anterior insula may suppress switching between these networks, leading to excessive self-reflection and enhanced awareness of self-signals which were previously filtered out of consciousness. Such features are indistinguishable from phenomenological descriptions of SD [45]. Several studies go further, arguing that hyperactivation of the insula leads to enhanced awareness of self-signals [60], a core feature of SD. Other studies show that self-referential activities specifically activate the insular cortex. For example, participants engaged in a heartbeat detection task where they were required to judge whether tonal feedback was synchronous with a heartbeat or delayed achieved greater accuracy through greater activation of the insula [77]. Furthermore, during a source misattribution task where participants were tasked with discriminating between intact and altered pictures of their face and body or that of a close colleague, fMRI imaging showed different areas (foci) of the anterior insula were activated when viewing one’s own face versus that of one’s colleague [78].

Research elsewhere in psychiatry suggests hyperactivation of Von Economo Neurons (VENs) may underlie such network dysfunction. VENs are only present within the anterior insula and anterior cingulate cortex and post mortem studies have found greater lysosomal aggregations in the VENs of those with schizophrenia, and to a lesser degree bi-polar disorder, suggesting a vulnerability to VEN damage may be associated with psychotic disorders [79]. VENs are also more densely populated in the right anterior insula compared to the left anterior insula, and are more closely tied to self-related cognition [80] and negative emotion [81]. For example, although the mechanism remains unclear, VEN activation in the right anterior insula appears to produce an increase in error processing and feelings of uncertainty [82], both common characteristics of SD. Similarly, overactive or over-abundant VENs are reported in both schizophrenia and autism and may be associated with heightened interoception [79,81,83]. In children, self-control is related to left-hemisphere activation of VENs, whereas, in adolescents, self-control is associated with right-hemispheric VENs [84]. The authors argue that the demands of adolescence require greater error monitoring due to the increased risk of social and physical harms, which leads to a shift towards right-hemisphere VEN activation. These findings may explain why the onset of SD is often reported in late childhood or early adolescence, assuming underlying neurodevelopmental abnormalities alone or in combination with traumatic experiences [84]. We hypothesize that excessive activation of the SN, as a result of neurodevelopmental abnormalities or exposure to trauma, may be associated with a state of hyper-reflexivity and a greater focus towards the self [78,80], and may lead to a lack of suppression of the DMN during cognitively demanding tasks. As a result, an increased awareness of previously filtered out self-signals becomes present. Based on these findings, SD may be explained by triple network theory. As such, we propose a triple-network model of SD.

### 4.5. A Triple Network Model of Self-Disorder

The literature reviewed for this article describes a possible dysfunction of three distinct brain networks. Within a triple-network model of SD, hyperactivation of the salience network may be linked to the following components: increased inter-connectivity between the SN and the FPN, decreased inter-connectivity between the FPN and DMN, increased inter-connectivity between the SN and DMN, and increased intra-connectivity within both the DMN and FPN (See model in Appendix A).

### 4.6. Limitations

There are several notable limitations in the methodology of included studies that need to be addressed in future research to test the assumptions of our model. Firstly, we did not identify any studies containing the relationships between neurotransmission and SD using our search strategy. Secondly, there is variability in the quality of outcome measures of SD in question across seven of the studies, six of these due to administering the BSABS 82 [39,41,43,50,52,54] and the remaining study for utilizing only a single item of the EASE rather than total and domain composites [42]. Although substantial overlap exists between certain subscales of the BSABS and SD [25], the scale was not designed for the purpose of assessing SD. Similarly, a single item of the EASE is likely to provide a poor marker for SD [85,86], as singular EASE items are more likely to be endorsed by controls or those experiencing comorbid psychopathology [25]. Thirdly, science now recognizes that experimental results from males may not be identical to females and vice versa, yet the studies included in this review did not report sex differences; therefore, these differences cannot be inferred. Fourth, small sample size is a recurring limitation across studies, particularly when such diverse methodology is employed. Although investigators have utilized both electrophysiological and imaging techniques, particularly EEG and fMRI, results from imaging studies were mixed. Finally, electrophysiology has distinct limitations when used in the absence of imaging techniques. Fundamentally, there are limitations to spatial resolution and connectivity analysis of EEG data limiting the accuracy of localization of the source of activation sites across the brain or determining whether activation is occurring at the cortical surface or at deeper layers [35]. For example, although midline nodes may represent midline structures of the DMN, it is difficult to eliminate noise within the system, such as that transmitted from neighboring nodes [62]. Such noise can often distort findings in EEG studies. As a result, although triple network theory provides an attractive conceptual framework for a neurophenomenology of Self-Disorder, one must be cautious when attempting to derive a spatial model of psychopathology from predominantly EEG data.

Available imaging studies report mixed findings; however, one structural MRI study reported a negative correlation between EASE and grey matter volume within the anterior cingulate cortex, a component of the SN [40], yet no significant structural differences in midline regions associated with the DMN. Although loss of grey matter volume may be interpreted as a sign of reduced function due to a reduction in cell size or number, structural differences say little about the activity of these cells. In those with schizophrenia, a loss of insula grey matter volume may also be associated with dysfunctional inhibitory signaling received from FPN regions such as the dorsolateral pre-frontal cortex, which moderates DMN suppression [87]. It is also important to note that grey matter volume fails to delineate pyramidal cells from VENs, and the ratio of VENs to pyramidal cells may be more important than mere volume [79]. For example, within autistic populations, findings indicate both high and low VEN density are related to dysfunction, with controls falling in the middle of the distribution [81]. Findings elsewhere may also resolve these conflicts.

Fundamentally, the triple network model of SD aims to describe a unique phenomenological experience of self-detachment: a loss of immersion in or grip on the world that could easily be described as a hypo-real state. In contrast, other studies find that both hyper- and hypoactivation of SN structures predict distinct syndromal subtypes of Post-Traumatic Stress Disorder [73,74]. Similarly, qualitative investigations into themes of delusional reality in psychotic states propose distinct sub-types of delusional experiences, each intimately tied to self-experience [88]. The authors found that half of participants described ‘hypo-real’ experiences of increasing detachment from self, uncertainty, hyper-reflectiveness, and doubt, a removal from the immersion of their actions and their participation with the world. In contrast, two thirds of participants described ‘hyper-real’ experiences of enhanced centrality of the self, meaningfulness, and significance in everyday experiences, where everything had a sense of necessity or compulsion as well as a loss of coincidence. Some displayed predominantly one form of reality, whereas a minority fluctuated between both extremes. Both groups were associated with aberrant salience and the authors went on to recommend further investigation to better understand the mechanisms underlying this divergence, mechanisms not captured within existing models of SD. Although findings support a triple network model of SD, further research is needed to better understand how physical disease or abnormal neurodevelopment can impact these networks to generate dysconnectivity and dysfunction.

### 4.7. Future Research Agendas Should Investigate Sub-Cortical Regions of Interest and Address Limitations by Building Translational Potential

To date, much of the investigation into the neurobiology of SD relies heavily on isolated electrophysiological signatures rather than taking a broader network approach. A triple-network model of SD is only one possible interpretation, though we believe such a model best-accommodates the physiological findings to date. However, a greater understanding of the nature of SD is needed and could be gained from thorough investigation into brain network connectivity between those regions of interest associated with the SN, DMN, and FPN. Imaging and EEG source localization techniques, such as e-LORETA, could be applied for this purpose across a range of psychiatric populations. Emerging evidence suggests a variety of techniques exist, some of which are easily accessible and, when used alongside fMRI, address limitations regarding localization accuracy and spatial resolution [89].

Despite the benefits of a more rigorous neurobiological research agenda, these findings fail to address the clinical utility of fMRI and EGG. Specifically, these modalities of assessment are expensive and time-consuming to both collect and analyze; as such, they are not available to the majority of clinicians and patients. Research could aim to develop novel phenomenological screening tools validated against neuroimaging and EEG that can be directly applied in clinical practice. In recent years, techniques such as facial and speech feature analysis have been developed for such a purpose [90]. Data can be gathered via telehealth with little participant discomfort, making these techniques easy-to-administer and ideal for use on a larger scale. Within the past decade, both methods have isolated a number of digital biomarkers for Major Depressive Disorder [91,92,93], Bipolar Disorder [94], and schizophrenia [95,96,97,98,99]. Within this review, correlations have been identified between facial mimicry and SD [50,100], suggesting some deficit in responding to facial expressions. Elsewhere, linguistic features were able to isolate features of SD within a schizophrenia population [101]. Further investigation of such novel methods may provide effective screening tools for SD within psychosis, especially if findings can be reconciled with the electrophysiological, anatomical, and neurocognitive characteristics described to date.

## 5. Conclusions

In this paper, we provided a systematic review of 21 studies exploring existing neurobiological signatures associated with Self-Disorder. Three themes were identified, predominantly taken from EEG and MRI studies, suggesting a dysfunction of brain network connectivity, abnormal oscillatory activity, and multi-model signal disintegration, which may contribute to Self-Disorder. These findings can be accommodated within a triple-network model in the context of triple network theory, which proposes that hyperactive insula function may lead to reduced inter-connectivity between the default mode network and the Fronto-Parietal Network, as well as increased intra-connectivity within both the default mode network and the Fronto-Parietal Network. We discuss findings in support of a triple-network model, address limitations, and suggest further investigation into network connectivity using source localization techniques as well as exploring novel screening tools, such as digital biomarkers, to address limitations in translational potential.

## Figures and Tables

**Figure 1 brainsci-13-00845-f001:**
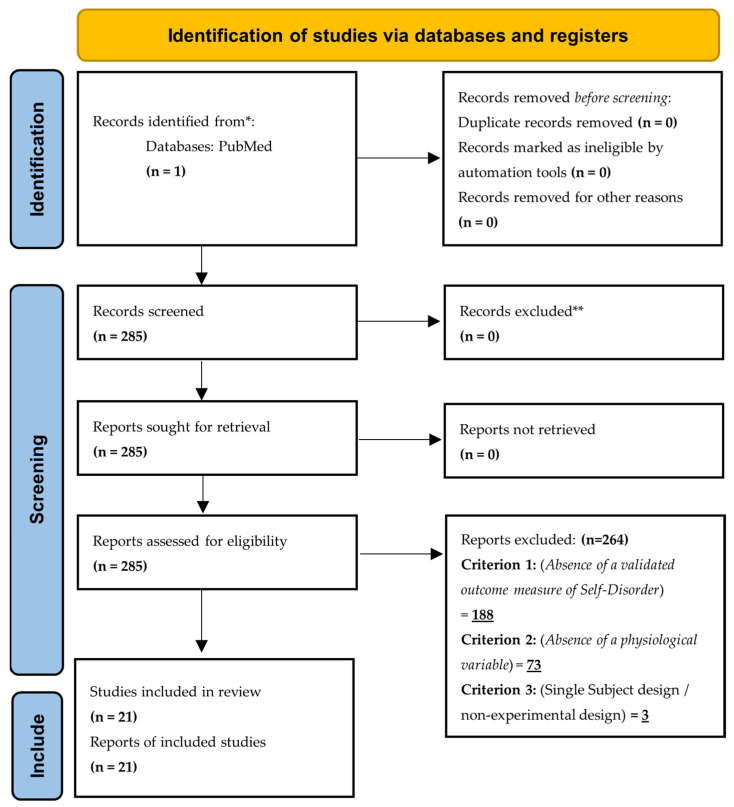
PRISMA flow chart of search strategy and screening. * = database name; ** = excluded before screening due to reasons listed.

**Table 1 brainsci-13-00845-t001:** Studies describing the association between measures of Self-Disorder and physiological measures of neural structure or function.

Name	N	Self-Disorder Measure (EASE, IPASE, BSABS)	Subcomponent of SD	Domain Measured Physiological Method	Effect Size/R Description of Results
Zhuo et al. (2021) [39]	30	BSABS	Total	**Grey matter volume** Magnetic Resonance Imaging	**Non-significant** No relationship was found between Basic symptoms and grey matter volume in midline cortical structures [39]
Bonoldi et al. (2019) [40]	47	EASE	Total	**Grey matter volume** Magnetic Resonance Imaging	**R = −0.52 *** Within the Ultra-High-Risk group, those with high EASE scores recorded smaller grey matter volume within the anterior cingulate cortex compared to those with low EASE scores [40]
Kéri et al. (2005) [41]	55	BSABS	Total	**Perceptual organization** EEG—Detection of gabor patches with collinear and orthogonal flankers	**R = 0.68 * (β = 0.75 ***)** Greater perceptual disorganization is predictive of higher BSABS scores and greater perceptual disorder [41]
				**Magnocellular (M) pathway** EEG—Low-contrast and frequency-doubling vernier threshold	**Low contrast (R = 0.65 *), frequency doubling (R = 0.53 *)** Deficits in the M pathway positively correlates with BSABS scores and greater perceptual disorder [41]
				**Parvocellular (P) pathway** EEG—Isoluminant color vernier threshold and high spatial frequency discrimination	**Non-significant** No relationship was found between Basic symptoms and changes in P pathway function [41]
Núñez et al. (2014) [42]	39	EASE	Distance to world factor	**Magnocellular and parvocellular pathways** EEG—Visual Evoked Potential paradigm	**N80 (F = 4.51 *)** Using an M priming task, greater N80 amplitude was associated with increased EASE scores on ‘distance to world’ items [42]
			World intrusion factor	**Magnocellular and parvocellular pathways** EEG—Visual Evoked Potential paradigm	**Non-significant** No relationship was found between M priming and EASE scores on ‘world intrusion’ items [42]
Brockhaus-Dumke et al. (2005) [43]	107	BSABS	Cognitive +	**Auditory sensory memory** EEG—left-frontal and fronto-central electrodes—mismatch negativity	**Non-significant** No relationship was found between auditory sensory memory impairment and scores using the BSABS-Cognition subscale [43]
Roig-Herrero et al. (2022) [44]	22	IPASE	Total	**Connectivity within DMN—right rACC and r-paraH** rs-fMRI	**R = 0.616 **** A relationship was found between greater self-reported symptoms of Self-Disorder and increased connectivity between the right anterior cingulate cortex and the right para-hippocampus [44]
				**Connectivity within DMN—right isthmus cingulate cortex and r-paraH** rs-fMRI	**R = 0.604 **** A relationship was found between greater self-reported symptoms of Self-Disorder and increased connectivity between the right isthmus cingulate cortex and the right para-hippocampus [44]
				**Connectivity within DMN—right precuneus cortex and r-paraH** rs-fMRI	**R = 0.443 *** A relationship was found between greater self-reported symptoms of Self-Disorder and increased connectivity between the right precuneus cortex and the right para-hippocampus [44]
				**Connectivity within DMN—left isthmus cingulate cortex and l-paraH** rs-fMRI	**R = 0.445 *** A relationship was found between greater self-reported symptoms of Self-Disorder and increased connectivity between the left precuneus cortex and the left para-hippocampus [44]
Northoff et al. (2021) [45]	73	EASE	Meditational relationship between Self-Disorder and negative symptoms	**Temporal integration** EEG-enfacement illusion	**Non-significant** [45]
Donati et al. (2021) [46]	10	EASE	self-awareness/presence	**Readiness potential—intentional binding** EEG—self-paced brisk fist-closure task	**RP slope (t = −0.87 ***)** Participants with reduced readiness potential displayed more prominent symptoms within the self-awareness domain of Self-Disorder [46]
				**Amplitude modulation of beta rhythms** EEG—self-paced brisk fist-closure task	**Beta ERS (t = −0.56 *)** Participants with weaker Event-Related Synchronization displayed more prominent symptoms within the self-awareness domain of Self-Disorder [46]
			Existential reorientation	**Amplitude modulation of beta rhythms** EEG—self-paced brisk fist-closure task	**Beta ERS (t = −0.57 *)** Participants with weaker Event-Related Synchronization displayed more prominent symptoms within the existential reorientation domain of Self-Disorder [46]
			Total	**Readiness potential—intentional binding** EEG—self-paced brisk fist-closure task	**RP slope (t = −0.64 **)** Participants with reduced readiness potential displayed more prominent symptoms of Self-Disorder [46]
Arnfred et al. (2015) [47]	12	EASE	Total	**Proprioception: gamma frequency** EEG—contralateral proprioceptive evoked oscillatory activity	**(*r* = −0.76 **)** Greater symptoms of Self-Disorder were associated with lower peak parietal gamma frequencies over frontal and parietal electrodes in the left hemisphere following right-hand proprioceptive stimulation [47]
				**Frontal beta amplitude** EEG—contralateral proprioceptive evoked oscillatory activity	**(*r* = 0.684 **)** Greater symptoms of Self-Disorder were associated with higher peak beta amplitude over frontal electrodes in the left hemisphere following right-hand proprioceptive stimulation [47]
				**Parietal beta amplitude** EEG—contralateral proprioceptive evoked oscillatory activity	**(*r* = 0.572 *)** Greater symptoms of Self-Disorder were associated with higher peak beta amplitude over parietal [47] electrodes in the left hemisphere following right-hand proprioceptive stimulation [47]
Hernández-García et al. (2020) [48]	25	IPASE	Total	**Connectivity strength** EEG—oddball paradigm pre-stimulus [PS], modulation [M]	**PS *(p* = 0.43 *), M (*p* = −0.4 **)*** Greater self-reported symptoms of Self-Disorder were positively associated with connectivity strength pre-stimulus but negatively associated with its modulation during a P300 task [48]
				**Spectral entropy—irregularity of signal** EEG—oddball paradigm—pre-stimulus	***P* = 0.41 *** A spectral entropy [SE] modulation deficit was associated with greater symptoms of Self-Disorder [48]
Martin et al. (2017) [49]	23	EASE	Total	**Reaction time** EEG—variable foreperiod paradigm	***r* = −0.4 *** Greater symptoms of Self-Disorder within the self-awareness domain were associated with reduced RT slope in the 0% catch trials condition [49]
				**Reaction time** EEG—variable foreperiod paradigm	***r* = 0.6 **** Greater symptoms of Self-Disorder within the self-awareness domain were associated with the change in RT slope for trials that followed a catch trial vs. those that followed a target-present trial [49]
Sestito et al. (2015) [50]	18	BSABS	Self-Disorder Subscale	**Congruent facial mimicry** Electromyography	**F (6,11) = 5.83 **, R^2^ = 0.76** Greater symptoms of Self-Disorder within the BSABS were associated with multi-modal deficits in congruent facial mimicry, suggesting deficits in emotional motor resonance to positive stimuli and excessive resonance to negative stimuli [50]

* *p* < 0.05, ** *p* < 0.01, *** *p* < 0.001, + substantial limitations in methodology, EASE: Examination of Anomalous Self-Experience, IPASE: Inventory of Psychotic-like Anomalous Self-Experience, BSABS: Bonn Scale for the Assessment of Basic Symptoms, EEG: Electroencephalography, DMN: Default Mode Network, rs-fMRI: resting-state functional Magnetic Resonance Imaging, BACS: Brief Assessment of Cognition in Schizophrenia, BetaERS: Beta Event-Related Synchronization, rACC: right anterior cingulate cortex, rParaH: right para-hippocampus.

**Table 2 brainsci-13-00845-t002:** Studies describing the association between measures of Self-Disorder and measures of perception.

Name	N	Self-Disorder Measure (EASE, IPASE, BSABS)	Subcomponent of SD	Domain Measured Physiological Method	Effect Size/R Description of Results
Nelson et al. (2020) [51]	123	EASE	Total	**Source monitoring** EEG—action memory task, word recognition test, temporal binding task	***r*^2^ = 0.41, F (13,85) = 14.78 ***** Source monitoring deficits explained 39.8% of variance across EASE scores, with greater source monitoring deficits predicting greater symptoms of Self-Disorder [51]
				**Source monitoring** EEG—auditory button press task—N1	**N1 suppression = −0.489 *** Greater symptoms of Self-Disorder were associated with less N1 suppression in First Episode Psychosis but not the Ultra-High-Risk group [51]
				**Aberrant salience** Salience attribution test, babble task	**non-significant** [51]
				**Aberrant salience** EEG—auditory oddball paradigm	**non-significant** [51]
Szily et al. (2009) [52]	68	BSABS	Total	**Recognition of cognitive expression** Reading the mind in the eyes test	***r* = −0.56 *** In the high-risk group, greater symptoms of Self-Disorder were associated with impaired recognition of facial expressions within the cognitive domain [52]
				**Recognition of social positive expression** Reading the mind in the eyes test	***r* = −0.4 *** In the high-risk group, greater symptoms of Self-Disorder were associated with impaired recognition of facial expressions within the social positive domain [52]
				**Recognition of social negative expression** Reading the mind in the eyes test	***r* = −0.37 *** In the high-risk group, greater symptoms of Self-Disorder were associated with impaired recognition of facial expressions within the social negative domain [52]

* *p* < 0.05, *** *p* < 0.001, EASE: Examination of Anomalous Self-Experience, BSABS: Bonn Scale for the Assessment of Basic Symptoms, EEG: Electroencephalography.

**Table 3 brainsci-13-00845-t003:** Studies describing the association between measures of Self-Disorder and measures of cognition.

Name	N	Self-Disorder Measure (EASE, IPASE, BSABS)	Subcomponent of SD	Domain Measured Physiological Method	Effect Size/R Description of Results
Kéri et al. (2005) [41]	55	BSABS	Total	**Intelligence** WAIS	**non-significant** [41]
				**Sustained attention** Continuous Performance Test	**non-significant** [41]
				**Visual processing speed** Categorization of briefly presented natural scenes	**R = −0.43 *** A relationship was found indicating greater self-reported basic symptoms is associated with decreased visual processing speed [41]
Hernández-García et al. (2021) [53]	41	IPASE	Consciousness	**Problem solving** BACS	***Z* = −2.31 *** Greater symptoms of Self-Disorder within the consciousness domain were associated with deficits in problem solving [53]
			Somatization	**Motor speed performance** BACS	***z* = −2.27 *** Greater symptoms of Self-Disorder within the somatization domain were associated with deficits in motor speed [53]
			Self-awareness and presence	**Motor speed performance** BACS	***z* = −3.28 ***** Greater symptoms of Self-Disorder within the self-awareness domain were associated with deficits in motor speed [53]
Rajender et al. (2009) [54]	70	BSABS	Total	**Body concept** Image-marking procedure	**Skewed small (*r* = 0.45 **),* Skewed large (*r* = 0.52 ***)** Cenesthesias were found to correlate positively with disturbances in body concept, including feeling as if body parts were unusually small [*skewed small*] or unusually large [*skewed large*] [54]
Brockhaus-Dumke et al. (2005) [43]	107	BSABS	Cognitive +	**Verbal executive function** Verbal fluency test	**F = 3.569 *** Participants displaying more ‘basic symptoms’ within the cognitive domain, scored lower on a test of verbal executive functioning [43]
				**Executive function** Wisconsin card sorting test	**F = 4.377 ***** Participants displaying more ‘basic symptoms’ within the cognitive domain, scored lower on a test of executive functioning [43]
				**Verbal intelligence** Multiple choice vocabulary test	**F = 3.532 *** Participants displaying more ‘basic symptoms’ within the cognitive domain, scored lower on a test of verbal intelligence [43]
Sandsten et al. (2022) [55]	70	EASE	Total	**Intelligence** CANTAB	**non-significant** [55]
Haug et al. (2012) [56]	57	EASE	Total	**Verbal memory** WMS-III	***r* = −0.316 *** (Greater symptoms of Self-Disorder were associated with lower scores on a measure of verbal memory) [56]
Trask et al. (2021) [57]	82	IPASE	Cognition	**Total cognition** MATRICS consensus cognitive battery (MCCB)	***r* = −0.325 *** Greater symptoms of Self-Disorder within the cognition domain were associated with lower scores on total cognition, as scored by the MCCB [57]
				**Attention** MATRICS consensus cognitive battery (MCCB)	***r* = −0.35 *** Greater symptoms of Self-Disorder within the cognition domain were associated with lower scores in a test of attention, as scored by the MCCB [57]
				**Visual learning** MATRICS consensus cognitive battery (MCCB)	***r* = −0.29 *** Greater symptoms of Self-Disorder within the cognition domain were associated with lower scores in a test of visual learning, as scored by the MCCB [57]
				**Reasoning** MATRICS consensus cognitive battery (MCCB)	***r* = −0.31 *** Greater symptoms of Self-Disorder within the cognition domain were associated with lower scores in a test of reasoning, as scored by the MCCB [57]
				**Working memory** MATRICS consensus cognitive battery (MCCB)	***r* = −0.29 *** Greater symptoms of Self-Disorder within the cognition domain were associated with lower scores in a test of working memory, as scored by the MCCB [57]

* *p* < 0.05, *** *p* < 0.001, + substantial limitations in methodology, EASE: Examination of Anomalous Self-Experience, IPASE: Inventory of Psychotic-like Anomalous Self-Experience, BSABS: Bonn Scale for the Assessment of Basic Symptoms, WAIS: Weschler Adult Intelligence Scale, BACS: Brief Assessment of Cognition in Schizophrenia, ROCF: Rey–Osterrieth Complex Figure, CANTAB: Cambridge Neuropsychological Test Automated Battery, WMS-III: Weschler Memory Scale (Third Edition), MCCB: MATRICS consensus cognitive battery.

## Data Availability

All data are published in available peer review journals.

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
