# Peer review of "Towards a Neurophenomenological Understanding of Self-Disorder in Schizophrenia Spectrum Disorders: A Systematic Review and Synthesis of Anatomical, Physiological, and Neurocognitive Findings"

_brainsci, 2023, doi:10.3390/brainsci13060845_

Round 1

Reviewer 1 Report

This paper aimed at providing a systematic review of neurophenomenological markers of self-disturbances in schizophrenia. The goal is to argue that  a “comprehensive neurophenomenological understanding of Self-Disorder may improve diagnostic and therapeutic practice. This systematic review aims to evaluate anatomical, physiological, and neurocognitive correlates of Self Disorder (SD), considered a core feature of Schizophrenia Spectrum Disorders (SSDs), and integrate findings towards a neuro-phenomenological understanding”. 

I believe the paper may deliver a good contribution to the field provided the Authors address satisfactorily  the points listed below.

Title: the term “neurophenomenology” traditionally points to a well established field, and seminal work on this should be cited, and defined, especially if the title contains the term. 

See also more recent work by N. Depraz, etc. 

Introduction : I’d like to suggest to provide short definitions of the key terms, for example schizophrenia, even if this term is widely used in the literature. Another key term to be defined is “passivity phenomena” L 47. I am not sure I understand the example provide at L 53-55. The idea of quality of experience (i.e. phenomenology, the ‘how’) is pervasive of ANY experience, so there is a phenomenological side of HOW related to both these experiences (i.e. i)the nurse is moving my arm; and ii) it feels radically altered. This example is highly unclear and may benefit from a clear definition of passivity phenomena. 

Line 67 and ff – provide REF as this concept means very different things for different authors, ex. Fuchs versus Metzinger; 

Line 89 – “heightened awareness of gustatory” – this is very interesting but a REF needs to be added

Line 143 – This is a very questionable interpretation of the Active Inference framework. The Active Inference framework stipulates, quite on the contrary, that the survival of the self IS the most relevant, and hence processed “transparently” ( see Limanowski and Friston 2020; Ciaunica et. al 2021. This later paper cited in this work has a very clear description of how the minimal self is attenuated. But attenuated does not mean that it is not “pertinent to survival” (L143).

L164 – It is important not to conflate qualitative methods such as EASE with neurophenomenology. 

L 168 – what is an “endophenotype”?

L 270 – 271 – I believe that the claim here is too ambitious. Listing studies on neural mechanisms on the one hand, and studies on qualitative phenomenology on the other hand, does not deliver the “integrated neurophenomenological model of self-disturbance”. It is important indeed to have this review and this picture, but integrating these findings into a coherent framework is way too ambitious for this paper, and this model is not delivered. For example, in the Figure 2, it is unclear how the inhibitory versus the enhanced connectivity work, given that all the arrow are the same. I’m afraid this Figure is highly unclear to me. Especially given that, as the Authors acknowledge, the studies reviewed are fairly few (21) and that most of them they have small sample sizes, and use just one EASE item out of the 39. With such a small sample it becomes very difficult to provide a network model of self-disorders. 

Overall, this is a very interesting paper and I applaud their authors for their hard work on this. 

Author Response

Dear reviewer,

Firstly, we thank you for your timely and detailed feedback. Please see below our responses to each of the comments made.

Reviewer 1

Comment 1: Title: the term “neurophenomenology” traditionally points to a well-established field, and seminal work on this should be cited, and defined, especially if the title contains the term. 

Reply: We note our work refers to a well-established field as indicated, and as such have provided a relevant citation and definition in the amended manuscript. Our manuscript now reads: “These considerations urge further exploration and synthesis [12,13], to provide comprehensive diagnostic and explanatory models that more closely reflect first-person experience whilst recognizing their reciprocal relationship to underlying neurophysiology (Varela, 1996). Such a neurophenomenological approach (Varela, 1996) may provide improved understanding of underlying mechanisms of psychosis leading to novel avenues for treatment and improve shared understanding of patient experience promoting better engagement and adherence to treatment.”(1, 2).

Comment 2: Introduction : I’d like to suggest to provide short definitions of the key terms, for example schizophrenia, even if this term is widely used in the literature. Another key term to be defined is “passivity phenomena” L 47. I am not sure I understand the example provide at L 53-55. The idea of quality of experience (i.e. phenomenology, the ‘how’) is pervasive of ANY experience, so there is a phenomenological side of HOW related to both these experiences (i.e. i)the nurse is moving my arm; and ii) it feels radically altered. This example is highly unclear and may benefit from a clear definition of passivity phenomena. 

Reply: We thank the reviewer for this comment, as the content we aim to communicate is challenging in nature and we believe clarity is paramount. Firstly, we have amended our manuscript to define terms such as schizophrenia and passivity phenomena, which we agree provides greater clarity to the following example. Secondly, although we recognize both descriptions of passivity phenomena entail a specific phenomenological quality, our aim here is only to suggest that clinical focus has previously centred on the attribution of passivity (i.e. the belief about who is controlling movement) rather than the experience itself, the cognitive rather than the phenomenological component. We aim to argue that the experience is primary and the belief secondary, rather than the other way around.

The manuscript now reads: “For example, whilst schizophrenia is currently defined only by the presence of psychotic features such as hallucinations, delusions, and disorganized or negative symptoms (3), these symptoms were historically considered peripheral, instead its core was best characterized by a loss of the innermost self (4). In modern psychiatry, whilst clinicians are still taught to identify relevant phenomena as part of mental state examination, such as passivity phenomena where one believes one’s thoughts or actions are controlled externally, constructs such as ‘self’ are not well operationalized in current diagnostic criteria and exploration of phenomenological experience tends to focus almost exclusively on the content of experience [1, 9, 10].”

Comment 3: Line 67 and ff – provide REF as this concept means very different things for different authors, ex. Fuchs versus Metzinger; 

Reply: Thanks for identifying this, we have now added this reference to the end of the sentence. (1)

Comment 4: Line 89 – “heightened awareness of gustatory” – this is very interesting but a REF needs to be added

Reply: Thanks for identifying this, we have now added a relevant reference to the end of the sentence. Reference added: (5, 6)

Comment 5: Line 143 – This is a very questionable interpretation of the Active Inference framework. The Active Inference framework stipulates, quite on the contrary, that the survival of the self IS the most relevant, and hence processed “transparently” ( see Limanowski and Friston 2020; Ciaunica et. al 2021. This later paper cited in this work has a very clear description of how the minimal self is attenuated. But attenuated does not mean that it is not “pertinent to survival” (L143).

Reply: We appreciate your bringing our attention to this confusion. We agree that minimal self signals are pertinent to survival and agree our manuscript fails to reflect this. We were in fact aiming to suggest that the conscious awareness of such signals does not further aid survival. Due to their highly predictable nature one can afford not to attend to them. As suggested by Ciaunica et al. (2021), this allows us to focus on more salient events in our proximal environment, which was the intention of the phrase “pertinent to survival”. We have amended our manuscript to better reflect this. Our manuscript now reads: “As these signals are highly predictable, we can afford to process them outside conscious awareness.  Consequently, most people produce efference copies to suppress ‘minimal self’ signals [15, 29], such as gustatory processes, heart rate, and somatosensory pathways. This allows the self to remain transparent and fully immersed with the world, whilst also allowing attentional resources to be directed towards more salient stimuli (Ciaunica et al., 2021).”

Comment 6: L164 – It is important not to conflate qualitative methods such as EASE with neurophenomenology.

Reply: We agree that the inclusion of neurophenomenology may confuse the reader and detracts from the aim of the paper, this has been removed to aid clarity. Our manuscript now reads: “We aim to integrate current knowledge, aligning existing models of SD with the broader neurobiological and computational literature.”  

Comment 7: L 168 – what is an “endophenotype”?

Reply: Similarly, we agree that the introduction of unnecessary terminology may confuse the reader and detracts from the aim of the paper, this has been adjusted to aid clarity. Our manuscript now reads: “we provide a systematic review of the proposed associations between SD and neurophysiological and neurocognitive correlates”.

Comment 8: L 270 – 271 – I believe that the claim here is too ambitious. Listing studies on neural mechanisms on the one hand, and studies on qualitative phenomenology on the other hand, does not deliver the “integrated neurophenomenological model of self-disturbance”. It is important indeed to have this review and this picture, but integrating these findings into a coherent framework is way too ambitious for this paper, and this model is not delivered. For example, in the Figure 2, it is unclear how the inhibitory versus the enhanced connectivity work, given that all the arrow are the same. I’m afraid this Figure is highly unclear to me. Especially given that, as the Authors acknowledge, the studies reviewed are fairly few (21) and that most of them they have small sample sizes, and use just one EASE item out of the 39. With such a small sample it becomes very difficult to provide a network model of self-disorders. 

Reply: We recognize that the proposed model does not deliver an integrated neurophenomenological model of self-disturbance and thank the reviewer for identifying the need to adjust language throughout. To address this comment we have reviewed the manuscript and made adjustments to the following lines, with the aim of highlighting the limitations of the proposed model: L17, L21, L280-283, L305-306, L428-431.

Also, in order to provide greater clarity to the GIF in figure 2, we have amended L437-446, and L449. Our manuscript now reads: “A theoretical triple-network model of Self-Disorder (arrows represent interactions between brain networks): Hyper-vulnerable Von Economo Neurons within the right anterior insula (rAI) and dorsal Anterior Cingulate Cortex (dACC) become dysfunctional following situational stress or neurodevelopmental processes, leading to hyperactivation of rAI and dACC (Salience Network, SN). In turn, this may inhibit the connectivity between the DMN and FPN, whilst enhancing connectivity between SN and DMN, as well as SN and FPN. Local connectivity may increase within the DMN and FPN following the loss of distal connections, leading to further activation of SN (rAI and dACC). SN: involved in attributing salience to stimuli and integrating sensory and interoceptive signals; FPN: involved in working memory, problem solving, and attention; DMN: involved in self-reflection in internal action.”

Yours Sincerely, 

The authors

Reviewer 2 Report

The authors examine self-disorder in schizophrenic illnesses, and they examine 21 references, after a wide research of references. They correlate these symptoms with neurophysiological features, and they describe a coherence of self-disorder with three important networks. This is an important review, because my teachers in psychiatry at the university of Bonn discovered the BSABS. The English language is very good, and the references are very well chosen. In the scheme at the end of the manuscript, I recommend to explain the three networks in easy words. In my opinion, this is an excellent review.

Author Response

Dear Reviewer,

Firstly, we thank you for your timely and detailed feedback. Please see below our responses to each of the comments made.

Comment 1: The authors examine self-disorder in schizophrenic illnesses, and they examine 21 references, after a wide research of references. They correlate these symptoms with neurophysiological features, and they describe a coherence of self-disorder with three important networks. This is an important review, because my teachers in psychiatry at the university of Bonn discovered the BSABS. The English language is very good, and the references are very well chosen. In the scheme at the end of the manuscript, I recommend to explain the three networks in easy words. In my opinion, this is an excellent review.

Reply: We thank the reviewer for their positive feedback and acknowledge improved clarity of figure 2 may increase readability for the audience. As such, we have added additional text to footnote beneath figure 2. Our manuscript now reads: “A theoretical triple-network model of Self-Disorder (arrows represent interactions between brain networks): Hyper-vulnerable Von Economo Neurons within the right anterior insula (rAI) and dorsal Anterior Cingulate Cortex (dACC) become dysfunctional following situational stress or neurodevelopmental processes, leading to hyperactivation of rAI and dACC (Salience Network, SN). In turn, this may inhibit the connectivity between the DMN and FPN, whilst enhancing connectivity between SN and DMN, as well as SN and FPN. Local connectivity may increase within the DMN and FPN following the loss of distal connections, leading to further activation of SN (rAI and dACC). SN: involved in attributing salience to stimuli and integrating sensory and interoceptive signals; FPN: involved in working memory, problem solving, and attention; DMN: involved in self-reflection in internal action”.

Yours Sincerely, 

The authors

Reviewer 3 Report

The authors present a systematic review of studies focused on brain areas associated with self-disorder.

The topic is highly interesting for the field, the review is well organized and written, though some concerns should be discussed:

- The term "endophenotypes" should be better defined. Indeed, if the phenotype is evident due to signs/symptoms, then it should be not called "endo" as it refers to "hidden or not noticeable yet". This is clear for example according to the two-hit hypothesis, where the circuit derangement will manifest the pathology only if it will be followed a second cue, such as a trauma or a drug (see for example preclinical studies 10.5402/2012/451865 or 10.1038/s41593-019-0512-2)

- In the text it should be clearly stated that the analyzed studies do not consider neurotrasmitters and/or neuronal subpopulations. Indeed, it is well recognized that neuropsychiatric disruptions relies beyond the overall brain regions or circuits that can be analyzed by EEG or fMRI. 

-Finally, gender of patients should be reported, since it is nowadays ackwnowledged that results from males could be not simply indicative for females, and viceversa.

Author Response

Dear Reviewer, 

Comment 1: - The term "endophenotypes" should be better defined. Indeed, if the phenotype is evident due to signs/symptoms, then it should be not called "endo" as it refers to "hidden or not noticeable yet". This is clear for example according to the two-hit hypothesis, where the circuit derangement will manifest the pathology only if it will be followed a second cue, such as a trauma or a drug (see for example preclinical studies 10.5402/2012/451865 or 10.1038/s41593-019-0512-2)

Reply: We appreciate this feedback and acknowledge that the introduction of unnecessary terminology such as this may confuse the reader and detracts from the aim of the paper, this has been adjusted to aid clarity. The manuscript now reads: “we provide a systematic review of the proposed associations between SD and neurophysiological and neurocognitive correlates”.

Comment 2: - In the text it should be clearly stated that the analyzed studies do not consider neurotrasmitters and/or neuronal subpopulations. Indeed, it is well recognized that neuropsychiatric disruptions relies beyond the overall brain regions or circuits that can be analyzed by EEG or fMRI. 

Reply:  We note there is a lack of information at the level of neuronal subpopulations and neurotransmission. The only evidence below the level of broad brain networks involves magnocellular pathways and this has been included in the review. We have amended our manuscript to list this within the limitations of our study. Our manuscript now reads: “we did not identify any studies containing the relationships between neurotransmission and SD using our search strategy.”

Comment 3: -Finally, gender of patients should be reported, since it is nowadays acknowledged that results from males could be not simply indicative for females, and vice versa.

Reply: Gender results were not reported within individual studies. As such, we are unable to list these differences. We do however acknowledge this as a limitation of the study and have now listed it as such. The manuscript now reads: “Thirdly, science now recognizes that experimental results from males may not be identical to females and vice versa, yet the studies included in this review did not report sex differences, therefore these differences cannot be inferred.”

Yours Sincerely, 

The authors